# A Novel Multiscale Methodology for Simulating Droplet Morphology Evolution during Injection Molding of Polymer Blends

**DOI:** 10.3390/polym13010133

**Published:** 2020-12-30

**Authors:** Lin Deng, Suo Fan, Yun Zhang, Zhigao Huang, Shaofei Jiang, Jiquan Li, Huamin Zhou

**Affiliations:** 1College of Mechanical Engineering, Zhejiang University of Technology, Hangzhou 310014, China; jsf75@zjut.edu.cn (S.J.); lijq@zjut.edu.cn (J.L.); 2School of Mechanical and Electrical Engineering, Wuhan Institute of Technology, Wuhan 430074, China; fan_suo@163.com; 3State Key Laboratory of Material Processing and Die & Mold Technology, Huazhong University of Science and Technology, Wuhan 430074, China; marblezy@hust.edu.cn (Y.Z.); huangzhigao@hust.edu.cn (Z.H.); hmzhou@hust.edu.cn (H.Z.)

**Keywords:** polymer blends, injection molding, droplet morphology, lattice Boltzmann method, multi-scale modeling

## Abstract

The morphology of polymer blends plays a critical role in determining the properties of the blends and performance of resulting injection-molded parts. However, it is currently impossible to predict the morphology evolution during injection molding and the final micro-structure of the molded parts, as the existing models for the morphology evolution of polymer blends are still limited to a few simple flow fields. To fill this gap, this paper proposed a novel model for droplet morphology evolution during the mold filling process of polymer blends by coupling the models on macro- and meso-scales. The proposed model was verified by the injection molding experiment of PP/POE blends. The predicted curve of mold cavity pressure during filling process agreed precisely with the data of the corresponding pressure sensors. On the other hand, the model successfully tracked the moving trajectory and simulated morphology evolution of the droplets during the mold-filling process. After mold-filling ended, the simulation results of the final morphology of the droplets were consistent with the observations of the scanning electron microscope (SEM) experiment. Moreover, this study revealed the underlying mechanism of the droplet morphology evolution through the force analysis on the droplet. It is validated that the present model is a qualified tool for simulating the morphology evolution of polymer blends during injection molding and predicting the final microstructure of the products.

## 1. Introduction

Polymer blends are ubiquitously used in machinery, electronics, transportation and our everyday life due to their enhanced properties such as light weight, high strength, and chemical resistance [1,2,3]. For the large unfavorable enthalpy of mixing, most physical blends of different polymers prove to be immiscible. The resultant small-scale arrangement of the phases constitutes the microstructure inside the blends. Microstructure is determined by comments properties, flow history and fractions. If the fractions of the components are sufficiently different, the sea-island-like microstructure will emerge in the blends where numerous droplets are immersed in the matrix [4]. Injection molding is a widely-used processing method in manufacturing polymer blends products. With the blend melt filling the mold cavity on macroscale, at the same time the droplets morphologies undergo drastic change on the mesoscale. The microstructure of the polymer blend has a significant effect on the mechanical, thermal, optical and physical properties of polymer blends and their injection-molded parts [5]. The latest researches showed that the orientation, alignment and size distribution of droplets can greatly increase the stiffness, strength and toughness of the blends [6,7]. The refractive index of the polymer blends on the orientation and vertical direction can differs by one or more orders of magnitude, resulting in harmful birefringence [8]. Therefore, it is crucial to accurately simulate the morphology evolution of polymer blends during injection molding, so as to predict and control the properties and performance of the molded products.

The existing researches on morphology evolution of polymer blends are still limited to simple flow fields. Both the phenomenological and theoretical models can be roughly divided into three categories: interface models based on Doi-Otha theory, component concentration models based on the Cahn-Hilliard equation and ellipsoid droplet models based on ellipsoidal droplet approximation. Doi and Ohta et al. used the specific surface area and interfacial tensor of the dispersed phase as the state variables, the time derivative of the variables were decomposed as the sum of the deformation and relaxation of the droplet due to the external flow and interfacial tension [9]. Lee and Park phenomenologically considered the relaxation mechanisms of the interfaces in heterogeneous systems and proposed a more general constitutive equation [10]. Lacroix and Grmela investigated the structural changes of the PP/(EVA–EMA) blends in transient shear flow using a modified version of the Grmela and Ait-Kadi models [11], and obtained description of these transient rheological data [12]. The celebrated Cahn–Hilliard (C–H) equation was a mathematical model of the process of phase separation in binary blends. Voit and Krekhov et al. [13] reproduced the essential spatial and temporal features of the composition patterns of PDMS/PEMS blends and agreed well with the Soret experiments [14]. Pincus et al. made an investigation into the kinetics of spinodal decomposition of rapidly quenched marginally incompatible polymer blends based on an extend C-H equation [15]. Prusty and Keestra et al. revealed the coarsening kinetics in PMMA/SAN28 blends with small-angle light scattering (SALS) and computationally with a diffuse-interface model based on the Cahn–Hilliard equation [16]. Tabatabaei Yazdi and Chan et al. studied the effect of different temperature gradient values on the surface enrichment rate of an unstable binary polymer blend using the nonlinear Cahn–Hilliard equation [17], coupled with the Flory–Huggins–de-Gennes theory [18]. The droplet models assumed the droplets shape obey the ellipsoid assumption with whatever deformation. Maffettone and Minale presented a simple phenomenological model (M-M model) for the deformation of a droplet immersed in a fluid subjected to a flow field with a uniform, but otherwise arbitrary, velocity gradient [19]. The M-M model degenerates into Taylor theory in the limit of slow flows as well as for high viscosity ratios. Wetzel and Tucker presented an analytical model for the deformation of an ellipsoidal Newtonian droplet [20], suspended in another Newtonian fluid with different viscosity and zero interfacial tension. The W-T model was reported exact for any linear velocity field and was not limited to small deformations. Jackson and Tucker developed a model to predict the transient shape evolution of an ellipsoidal Newtonian droplet with interfacial tension, suspended in another Newtonian fluid with a different viscosity using the Eshelby equivalent inclusion theory [21]. Yu and Zhou integrated the interfacial constitutive equation, so-called as the Boussinesq–Scriven equation [22], in the perturbation analysis on the flow field inside and outside the droplet and thus suggested a theoretical model for droplet dynamics and rheology of blends. Grmela and Bousmina treated the morphology both at local (Doi-Ohta type) and more macroscopic (droplet-like) scales and proposed a family of new models, making a direct link between flow and structure for immiscible mixtures of viscoelastic fluids undergoing high deformation flows [23]. Comprehensive reviews on these issues have been given by Minale [24], Puyvelde [25] and Fortelny [26].

The biggest difficulty in simulating the morphology evolution of polymer blends during injection molding lies in that the mold-filling flow of polymer blends is far more complicated than simple ideal flow and the scales of polymer melt and droplet morphology is significantly distinct and interactive [27]. The macroscopic polymer melt flow generates the external force field of the droplets morphologies evolution on the mesoscale. In turn, the statistical average of droplets morphologies determines the local rheology of the polymer blends on macroscale. Coupling the models on different scales of polymer melt flow and droplet morphology evolution has been one of the long-standing challenges in multiscale modeling of polymer dynamics for years [28].

It is feasible to analyze the morphology evolution process of polymer blends during injection molding using a macro- and meso-scale coupled model. The finite volume method (FVM) was used in the proposed model to simulate the polymer melt flow on the macroscopic scale [29,30]. The lattice Boltzmann method (LBM) was employed to describe the complex morphology evolution of droplets on the mesoscopic scale. In recent years, the LBM has gained much success in studying multiphase flows, i.e., emulsion flow, bubble flow and binary flow [31,32]. Its fundamental idea is to construct simplified kinetic models that incorporate the essential physics of microscopic processes [33]. Macroscopic hydrodynamic behaviors, such as interface dynamics, naturally emerge as a result of this kinetics. This paper proposed a novel coupling framework, consisting of the droplet trajectory tracking method and the phenomenological constitutive model of polymer blends. Given the macro- and meso-scales coupled, a multiscale model for simulating the droplet morphology evolution during the mold filling process has been established.

This paper is organized as follows. Section 2 detailed the coupled macro- and meso-scale model including the FVM-based model for mold filling flow of polymer blends melt, the LBM-based model for droplet morphology evolution and a bi-directional coupling framework between them. To validate the proposed model, the simulation results were compared with two injection molding experiments in Section 3. One is the mold cavity pressure monitoring experiment and the other is the SEM imaging of the droplet morphologies of the PP/POE part. The conclusions drawn from this paper and the outlook for future research were briefed in Section 4.

## 2. Multiscale Model

The proposed multiscale approach was the integration of macroscopic LBM and mesoscopic FVM. The macro- and meso-scale models were coupled in a novel way that the macroscopic flow simulation defined the exterior conditions surrounding the droplets while the mesoscopic droplets morphologies contributed to the rheology of the polymer blends. This coupling approach was achieved through tracking the droplet trajectory following the polymer melt flow and the phenomenological constitutive model for molten polymer blends.

### 2.1. Macroscale Model for Mold Filling Flow of Polymer Melt

On the macroscale, the FVM with SIMPLE algorithm was used to simulate the mold filling flow of polymer blends melt. The governing equations of velocity, pressure and temperature inside the mold cavity could be written as:

Continuity equation:(1)∇⋅u=0

Momentum equation:(2)$$\rho \left(\frac{\partial u}{\partial t}+u\cdot \nabla u\right)=-\nabla p+\mu \nabla^{2}u+\rho g$$

Energy equation
(3)$$c_{p}\rho \left(\frac{\partial T}{\partial t}+u\cdot \nabla T\right)=\nabla \cdot \left(k\nabla T\right)+\dot{\Phi }$$

The variables and their meanings of the above equations were listed in Table 1 below:

As the mold filling process was a kind of free surface flow, the volume of fraction (VOF) method was employed to re-construct the melt-air interface advancement, and the transportation equation for the filled fraction went as follows:(4)$$\frac{\partial \chi }{\partial t}+\nabla \cdot \left(\chi u\right)=0$$
where χ was the filled fraction of control volume:(5)$$\chi \left(x,t\right)=\left\{ \begin{array}{l} 1,\;\mathrm{filled}\;\mathrm{with}\;\mathrm{polymer}\;\mathrm{melt}\\ 0,\;\mathrm{filled}\;\mathrm{with}\;\mathrm{air} \end{array}\right.$$

The conservative Equations (1)–(3) together with the transportation Equation (4) constituted the governing equations of the mold filling flow of the polymer blends melt.

The boundary conditions for the flow continuity and momentum equations must be prescribed to complete the mathematical description of the whole flow domain. The boundary conditions on the inlet and mold wall went as:(6)$$u=u_{inlet}\;\mathrm{on}\;\Gamma_{inlet},$$
(7)$$u=0\;\mathrm{on}\;\Gamma_{wall,filled}$$

At the flow front, interfaces between the melt and air, the traction free boundary condition and the atmospheric pressure condition were imposed.

For the energy equation, the initial conditions for the flow domain and the boundary conditions for the injection gate and mold wall were defined as follows:(8)$$T=T_{0}\left(x\right),\;\forall x\in \Omega ,\;t=0,$$
(9)$$T=T_{inlet}\left(t\right),\;\forall x\in \Gamma_{inlet},\;t>0,$$
(10)$$T=T_{wall}\left(t\right),\;\forall x\in \Gamma_{wall},\;t>0.$$

Besides, the initial and boundary conditions for χ of Equation (4) were:(11)χ=0, ∀x∈Ω, t=0,
(12)$$\chi =1,\;\forall x\in \Gamma_{inlet},\;t>0.$$

The governing Equations (1)–(4) can be written in the form of the following generalized transport equations
(13)$$\underset{\mathrm{transient}\;\mathrm{term}}{\frac{\partial \left(\Lambda \phi \right)}{\partial t}}+\underset{\mathrm{convective}\;\mathrm{term}}{\nabla \cdot \left(u\Lambda \phi \right)}-\underset{\mathrm{diffusive}\;\mathrm{term}}{\nabla \cdot \left(\Gamma \nabla \phi \right)}=\underset{\mathrm{source}\;\mathrm{term}}{Q_{\phi }}$$

The physical meanings of ϕ, Λ, Γ and $Q_{\phi }$ depend on the specific equation represented by Equation (13), as detailed in Table 2.

By integrating Equation (13) and applying the Gaussian divergence theorem, the discretized equation was obtained after some algebraic operations:(14)$$\Lambda_{P}V_{P}\frac{\phi_{P}-\phi_{P}^{0}}{\Delta t}+\sum_{f}\left(F_{f}^{c}-F_{f}^{d}\right)=V_{P}{\left(Q_{\phi }\right)}_{P}$$

Finally, the discrete equations of velocity, pressure and temperature were solved using the SIMPLE algorithm while the CICSAM method proposed by Ubbink and Issa et al. was utilized to maintain the sharpness of the melt-air interface and the boundedness of χ [34].

### 2.2. Mesoscale Droplet Morphology Evolution

On the mesoscale, the Shan-Chen pseudo-potential scheme of LBM [35,36] was employed to model the morphology evolution of the droplet immersed in the matrix.

Under the LBM framework, the continuous velocity space was discretized into a finite set of N vectors, and the density distribution function $f_{i}\left(x,t\right)$ denoted the probability of finding a particle with the discrete velocity $e_{i}$ at position x and time t. In this paper, the D3Q19 lattice model was used, as shown in Figure 1.

For the components of droplets and the matrix, they were represented by $f_{i}^{d}\left(x,t\right)$ and $f_{i}^{m}\left(x,t\right)$, respectively, and their governing equations were expressed as:(15)$$f_{i}^{\sigma }\left(x+e_{i}\Delta t,t+\Delta t\right)-f_{i}^{\sigma }\left(x,t\right)=\frac{1}{\tau^{\sigma }}\left[f_{i}^{\sigma }\left(x,t\right)-f_{i}^{\sigma ,eq}\left(x,t\right)\right]$$
where $f_{i}^{\sigma ,eq}\left(x,t\right)$ was the equilibrium distribution function corresponding to the component σ with velocity $e_{i}$ and was determined by:(16)$$f_{i}^{\sigma ,eq}=w_{i}\rho^{\sigma }\left(x,t\right)\left[1+3\left(e_{i}\cdot u^{\sigma ,eq}\right)+\frac{9}{2}{\left(e_{i}\cdot u^{\sigma .eq}\right)}^{2}-\frac{3}{2}\left(u^{\sigma ,eq}\cdot u^{\sigma ,eq}\right)\right]$$
with the weight coefficients:(17)$$w_{i}=\{\begin{array}{c} 1/3\;\;\;\;\;\;\;\;\;i=0\\ 1/18\;\;\;i=1-6\\ 1/36\;i=7-12 \end{array}$$

$\tau^{\sigma }$ was the dimensionless relaxation time and related to the kinematic viscosity of the component by $\nu^{\sigma }=\left(\tau^{\sigma }-0.5\right)/3$. $\rho^{\sigma }$ and $u^{\sigma }$ were the macroscopic density and velocity of component σ and calculated from the zeroth and first moments of $f_{i}^{\sigma }\left(x,t\right)$:(18)$$\rho^{\sigma }\left(x,t\right)=\sum_{i}f_{i}^{\sigma }\left(x,t\right)$$
(19)$$\rho^{\sigma }u^{\sigma }\left(x,t\right)=\sum_{i}f_{i}^{\sigma }\left(x,t\right)e_{i}$$

It should be noted that the expression of the equilibrium velocity $u^{\sigma ,eq}$ took into account the interaction from other components acted on component σ:(20)$$u^{\sigma ,eq}\left(x,t\right)=\frac{\sum_{\sigma }\rho^{\sigma }u^{\sigma }/\tau^{\sigma }}{\sum_{\sigma }\rho^{\sigma }/\tau^{\sigma }}+\frac{\tau^{\sigma }F^{\sigma }}{\rho^{\sigma }}$$

In the Shan-Chen pseudo-potential scheme, the total force on component σ results from the interaction of component $\bar{\sigma }$ at neighboring lattices, as illustrated in Figure 2.
(21)$$F^{\sigma }\left(x,t\right)=-G\rho^{\sigma }\left(x,t\right)\sum_{i}w_{i}\rho^{\bar{\sigma }}\left(x+e_{i}\Delta t,t\right)e_{i}$$
where G denotes the strength of the interaction and would determine the resultant interfacial tension as well as the interface thickness.

### 2.3. Macro- and Meso-Scale Coupling

#### 2.3.1. Droplet Trajectory Tracking

For the size of the droplets was far smaller than that of a control volume, when a droplet, marked as P, passed through a control volume, the droplet can be treated as a particle and its trajectory was tracked through first-order integral of its velocity. The droplet acceleration was obtained by the chain rule of derivative, i.e., in the *x* direction:(22)$${\left(dv_{x}/dt\right)}_{p}=\left(dv_{x}/dx\right){\left(dx/dt\right)}_{p}$$

Integrating the two sides of Equation (22) between time $t_{n}$ and $t_{n+1}$ ($t_{n+1}=t_{n}+\Delta t$) gave:(23)$$x_{p}\left(t_{n+1}\right)=x_{n}+\left(1/A_{x}\right)\left[v_{xp}\left(t_{n}\right)\exp\left(A_{x}\Delta t\right)-v_{xn}\right]$$

Using a similar derivation process, the coordinates of the droplet in the y-direction and the z-direction can also be updated from $t_{n}$ to $t_{n+1}$:(24)$$y_{p}\left(t_{n+1}\right)=y_{n}+\left(1/A_{y}\right)\left[v_{yp}\left(t_{n}\right)\exp\left(A_{y}\Delta t\right)-v_{yn}\right]$$
(25)$$z_{p}\left(t_{n+1}\right)=z_{n}+\left(1/A_{z}\right)\left[v_{zp}\left(t_{n}\right)\exp\left(A_{z}\Delta t\right)-v_{zn}\right]$$

The tracking operation was repeated in all control volumes where droplet *P* traversed one by one. During the mold filling flow, the droplets were largely subjected to the shear force of the flow field that depend on the location of the droplet. Given the droplet’s trajectory, the history of shear force on the droplet during mold filling was decided accordingly by interpolation, and then the exterior conditions for the morphology evolution of the droplets were well prepared.

#### 2.3.2. Model Set-Up for Droplet Morphology

In multiscale fluid dynamics and modeling, the complicated flow field on the macro scale always appeared as simple flow forms on the local mesoscale [37]. Macroscopically, the mold filling flow of polymer melt was a typical laminar flow [38,39,40,41], while mesoscopically the droplets immersed in the mold filling flow were mainly subjected to shear flow forces [42,43,44]. Based on this analysis, a local model of droplet morphology evolution on the mesoscale was set up accordingly.

As shown in Figure 3, a virtual shear flow field of time-varying shear rate $\dot{\gamma }\left(t\right)$ was produced by two oppositely-moving plates. An initially sphere droplet was placed in the center and the droplet morphology was about to change under the shear force.

The shear rate $\dot{\gamma }$ was not constant but calculated according to the location of the droplet in the flow filed. For instance, given the locations of the droplet at time $t_{n}$ and $t_{n+1}$ by tracking the droplet trajectory, then the values of shear rate and viscosity at time $t_{n}$ and $t_{n+1}$ was obtained from the macroscopic simulation. During the timestep, the shear rate on the droplet was assumed linearly varying from ${\dot{\gamma }}_{n}$ to ${\dot{\gamma }}_{n+1}$:(26)$$\begin{array}{l} \dot{\gamma }\left(t_{n}\right)=\dot{\gamma }\left(x_{n},y_{n},z_{n}\right)\\ \dot{\gamma }\left(t_{n+1}\right)=\dot{\gamma }\left(x_{n+1},y_{n+1},z_{n+1}\right) \end{array}$$
(27)$$\dot{\gamma }\left(t\right)=\frac{t_{n+1}-t}{t_{n+1}-t_{n}}\dot{\gamma }\left(t_{n}\right)+\frac{t-t_{n}}{t_{n+1}-t_{n}}\dot{\gamma }\left(t_{n+1}\right),\;t_{n}\leq t\leq t_{n+1}$$
where $\left(x_{n},y_{n},z_{n}\right)$ and $\left(x_{n+1},y_{n+1},z_{n+1}\right)$ were the coordinates of the droplet at time $t_{n}$ and $t_{n+1}$ separately.

The morphology evolution of the droplet during one timestep from $t_{n}$ to $t_{n+1}$ was simulated using the Shan-Chen pseudo-potential LBM, namely solving Equation (15).

#### 2.3.3. Polymer Blends Constitutive Equation

Compared with single-component polymers, the droplets morphologies directly contributed to the rheology of polymer blends and affected the macroscopic flow of the polymer blends. Yu quantified the effect of the interface structure on the local stress of the blends based on the ellipsoidal description of droplets [45]. However, the number density of droplets in polymer blends was extremely large, the morphologies of droplets were irregular and complex so that developing a theoretical constitutive model for polymer blends was nearly impossible. From another perspective, it is valuable to propose a phenomenological constitutive model without explicitly including the interface term. In this paper, the widely-used Cross-WLF equation, Equations (28) and (29), was utilized to calculate the viscosities of the polymer blends melt for the macroscopic simulation of the mold filling flow.
(28)$$\eta \left(T,\dot{\gamma }\right)=\frac{\eta_{0}\left(T,P\right)}{1+{\left(\frac{\eta_{0}\;\dot{\gamma }}{\dot{\tau }}\right)}^{1-\tilde{n}}}$$
(29)$$\eta_{0}=D_{1}\exp\left(\frac{-A_{1}\left(T-\left(D_{2}+D_{3}P\right)\right)}{A_{2}+T-D_{2}}\right)$$
where $\tilde{n}$ was the power law index in the high shear rate regime, $\dot{\tau }$ was the critical stress level at the transition to shear thinning, $A_{1}$, $A_{2}$, $D_{1}$, $D_{2}$ and $D_{3}$ were the material coefficients. All these coefficients in the Cross-WLF can be determined by fitting the rheological experiment data of the polymer blends.

### 2.4. Algorithm Summary

To summarize, the proposed multiscale model can be implemented with the following steps for the droplet during the mold filling process:(1)Given the initial macroscopic flow field of $u_{0}$, $p_{0}$ and $T_{0}$, the droplet morphology and coordinate $r_{0}$ at time $t_{0}$;(2)Calculate the shear rate on the droplet ${\dot{\gamma }}_{0}$ according to its coordinate $r_{0}$;(3)Solve Equations (1)–(4) to update the velocity, pressure and temperature to $u_{1}$, $p_{1}$ and $T_{1}$ to time $t_{1}=t_{0}+\Delta t$;(4)Update the coordinate of the droplet to $r_{1}$ at time $t_{1}$ according Equations (23)–(25);(5)Calculate the shear rate on the droplet ${\dot{\gamma }}_{1}$ according to its coordinate $r_{1}$;(6)Solve Equations (15) and (27) to simulate the droplet morphology evolution during the timestep between $t_{0}$ and $t_{1}$;(7)Go back to step (2) until the mold filling ends.

To make it clearer, the bidirectional coupled framework for the droplet morphology evolution during the mold filling of polymer blends was outlined in Figure 4 below.

## 3. Experimental Validation

As the proposed model spanned over macro- and meso-scales, two sets of injection molding experiments of polymer blends were conducted. Both sets of experiments were carried out in the HTL-90-F5B injection molding machine (produced by Ningbo Haitai Plastic Machinery Co., Ltd., Ningbo, China). The processing parameters were defined mainly according to the experience of the injection molding in consideration of the geometry, size and material of the molded product, assisted by the parameters provided by Moldflow Synergy as well. Meanwhile, corresponding simulations were carried out and the comparison between the experiments and simulations was made in order to validate the model.

### 3.1. Cavity Pressure Variation during Mold Filling

Cavity pressure prediction was the benchmark test for the macroscopic simulation of mold-filling flow. As shown in Figure 5, the molded part was a thin slat with the length, width and thickness of 220, 40 and 2.5 mm, respectively. Two pressure sensors were fixed on the surface of the mold cavity, one near the injection gate, P1, and the other one near the end, P2.

The processing parameters of this experiment was prescribed in Table 3.

The polymers used in the experiment were the blends of polypropylene (PP, trademark T36f, supplied by Sinopec Corp., Wuhan, China) and polyolefin elastomers (POE, trademark 8150, supplied by DowDuPont Inc., Wilmington, NC, USA).The major physical properties of PP and POE were listed in Table 4.

The granules of 25% POE and 75% PP were blended using a Haake (Vreden, Germany) twin-screw extruder machine. There are two reasons for the fixed fraction of POE. Firstly, this research was in the context of the manufacturing product of polymer blends. Trough mechanical testing of the products of various fraction, it is found that the molded parts of PP/POE blends with this fraction demonstrates the optimal mechanical performance and formability. Secondly, the fraction of components decides the morphology of the disperse phase and this paper was mainly focused on the sea-island microstructure.

Measured by the capillary rheometer, the shear viscosities versus shear rate of PP, POE and PP/POE blends at 210 °C were plotted in Figure 6.

Fitting the rheological data to the Cross-WLF viscosity model, seven model parameters of PP/POE blends were calculated and listed in Table 5.

Figure 7 compared the cavity pressure variation over time at P1 and P2 that were obtained numerically and experimentally. It can be seen from the comparison that the pressure transmission and rise at P1 was more complete and rapid than that of P2 because of different distances to the injection gate. The agreement between the experimental data and simulation results indicated that the macroscopic part of the present model was definitely validated.

### 3.2. Droplet Morphology Evolution during Mold Filing

#### 3.2.1. Experiments and Simulation Set-Up

Figure 8 showed the part for the observation and simulation of the droplet morphology evolution. For convenience of making brittle sections at the designated locations on the part, a smaller-sized (80 × 10 × 4 mm^3^) product was used, instead of the slat in the previous experiment. After molded, the parts would be frozen in liquid nitrogen for 24 h and then made brittle sections, and the solidified droplets morphologies were observed with the SEM.

Figure 9 depicted the locations and orientations of the brittle sections on the part. The sections were fabricated on two locations, (1) and (3), one adjacent to the injection gate (20 mm distant) and the other one far away from it (60 mm distant). To view the droplets morphologies from various angles, two sections were made on both locations, one section was parallel with the injection direction while the other one perpendicular to it.

Besides validating the droplet morphology simulation, the effect of injection rate on droplet morphology was investigated numerically. For this purpose, three cases (I, II and III) were implemented and injection rate was the single variable among them, as was marked in Table 6 below.

#### 3.2.2. Numerical Validation

Figure 10 showed the simulated trajectories and morphologies of the droplets of case II. Two droplets, marked A and B, were involved in the simulation. To better present the simulated process of droplets morphologies evolution, the droplets morphologies at location (2) were supplemented in addition to (1) and (3).

On droplets trajectories, it was obvious that the two droplets diverged almost shortly after they entered the mold cavity through the injection gate and subsequently flowed parallel to the injection direction. Droplet A drifted mainly in the shear layer, roughly 1.0 mm from the part surface, while droplet B mainly in the core layer, approximately 2.0 mm from the part surface. This trajectory pattern coincided with the laminar streamline of the mold-filling of polymer melt.

On droplets morphologies, the deformation, breakup and retraction of the droplets were figured out on their trajectories. Droplet A underwent drastic deformation and even broke up before the polymer flow ceased. In sharp contrast, droplet B was only slightly deformed throughout its trajectory and nearly retracted to a sphere finally.

Figure 11 and Figure 12 demonstrated the SEM images taken at locations (1) and (3) inside the injection molded part of case II respectively. Both sets of SEM images were focused on the shear layer, precisely 1.0 mm from the part surface, so they were compared with the simulation results of droplet A. From Figure 11b and Figure 12b, the cross-sectional profile of the droplets on the perpendicular section was approximately ellipses, suggesting that the droplets orientated at some an angle to the injection direction. By comparing Figure 11a and Figure 12a, it could be seen that the droplets were substantially deformed at location (1) and there were abundant evidence of droplet breakup and retraction at location (2). Both phenomenon on the SEM images agreed very well with the above simulations.

#### 3.2.3. Force Analysis of Droplets

Figure 13 showed the curve of shear stress and interfacial tension of the two droplets over time during the mold filling process. The combined action of shear stress and interfacial tension were key factors determining the droplets morphologies. Droplet A was subjected to greater shear stress than droplet B throughout mold filling. This led to larger deformation of droplet A than droplet B. On the other hand, according to Young-Laplace law, larger droplet deformation meant that the interfacial tension experienced by droplet A was also greater. With the weakening of shearing stress and the strengthening of interfacial tension, the deformation of the droplets gradually reached the maximum degree when shear stress and interfacial tension stroke a balance. Whether the droplets broke up or not, both droplets retracted more or less. The different interfacial tension and shear stress of the droplets, which was caused by their distinct trajectories in the flow field, explained the various morphology evolution of droplets.

#### 3.2.4. Effect of the Injection Rate

Figure 14 and Figure 15 showed the simulation results of the morphology evolution of droplet A along its trajectory under different injection rates (case I and III). Three snapshots of the droplets when they were at locations (1), (2) and (3) were presented. Compared with the droplet morphology evolution of case II, the droplet was more deformed and stretched into a worm-like shape under higher injection rate. Similarly, the droplet broke up more intensely and split into more than two sub-droplets of case II. It was noteworthy that at location (2) the middle sub-droplet even had a tendency of secondary breakup under higher injection rates. However, under lower injection rate, the droplet only deformed to a certain degree without any breakup.

Figure 16 and Figure 17 demonstrated the SEM images taken from the part of case I and III. These images were also focused on the shear layer at locations (1) and (3), specifically 1.0 mm from the surface, corresponding to droplet A of the simulation. The SEM experiments results revealed the impact of injection rate on droplet morphology which were consistent with the simulation. At location (1), the droplets were more deformed under higher injection rates than that under lower injection rates. The biggest difference between the droplets morphologies in Figure 16 and Figure 17 was at location (3), where larger amount of sub-droplets than that under lower injection rates, a sign of droplet breakup, were spotted in both SEM images. It suggested that higher injection rates significantly promoted deformation and breakup of droplets and agreed well with the simulation results (Figure 14c and Figure 15c).

Figure 18 illustrated the statistics of the droplets morphologies on Figure 16 and Figure 17. Here the droplets were assumed axisymmetric and ellipsoidal while the semi-major and semi-minor axes of the droplets, *L* and *W*, were measured on the SEM images using the image analysis program Nano measurer. Then the equivalent radius *R* and deformability *D* of the droplets were given by:(30)$$R=\sqrt[{3}]{LW^{2}}$$
(31)$$D=\frac{L-W}{L+W}$$

With the equivalent radius and deformability of all the droplets calculated, the average radius and deformability and their standard deviation of the droplets at position (1) and (3) of the parts molded under high and low injection rates were obtained using the following formula:(32)$$\bar{R}=\frac{1}{N}\sum_{i}^{N}R_{i},\;\bar{D}=\frac{1}{N}\sum_{i}^{N}D_{i}$$
(33)$$\sigma_{R}=\sqrt{\frac{\sum_{i}^{N}{\left(R_{i}-\bar{R}\right)}^{2}}{N}},\;\sigma_{D}=\sqrt{\frac{\sum_{i}^{N}{\left(D_{i}-\bar{D}\right)}^{2}}{N}}$$

From the statistics it is straightforward that higher injection rate resulted in smaller size and larger deformability of droplets on average, while along the trajectory from location (1) to (3) the averaged deformability decreased but the averaged radius increased. It is the trade-off result of droplet retraction and deformation. It also suggested that coalescence of droplets was promoted when the shear action weakened. The statistical results of the simulation further validated the proposed model.

## 4. Conclusions and Outlook

In this study, a novel model for the droplet morphology evolution during the injection molding of polymer blends was proposed using the multiscale approach of the FVM, LBM and local scale-coupling method. The macro- and meso-scales were successfully bridged by tracking droplet trajectory and developing the constitutive model of polymer blends. As the cornerstone of the multiscale model, the macroscopic simulation of the mold filling flow was verified by the mold cavity pressure experiment and the simulation results precisely agreed with the pressure sensors’ data. On the mesoscale, the model accurately tracked the trajectory of the droplet and simulated the subtle morphology evolution of the droplet on its trajectory. The simulation results and SEM experiments agreed very well so that the mesoscopic simulation of droplet morphology was validated. Through the numerical force analysis of the droplet, it is found that the shear stress sharply increased in the vicinity of the injection gate, resulting in abrupt deformation of the droplet. As the droplet moved away from the gate, the shear stress fell drastically while the interfacial tension gradually gained dominance, making the droplet tend to retract. Finally, the impact of injection rate on the droplet morphology evolution was investigated and the simulation results qualitatively agreed with the SEM observations, which further validated the proposed model.

As for the outlook for future research, there are two issue worthy of attention. Firstly, the POE fraction of the blends in this paper is fixed 25% for the sake of subsequent manufacturing quality. The impact of blends fraction on the morphology evolution could be further studied. Secondly, fountain flow is typical of injection molding of polymers. However, the morphology evolution of the droplets in the flow front is not shown in this paper due to the difficulty of experimental observation. To overcome this thorny problem, it requires novel experiment design.

In summary, this paper has achieved some pioneering success in multiscale simulation of injection molding of polymer blends, much work needs doing and it is firmly believed that this field is interesting and meaningful.

## Figures and Tables

**Figure 1 polymers-13-00133-f001:**
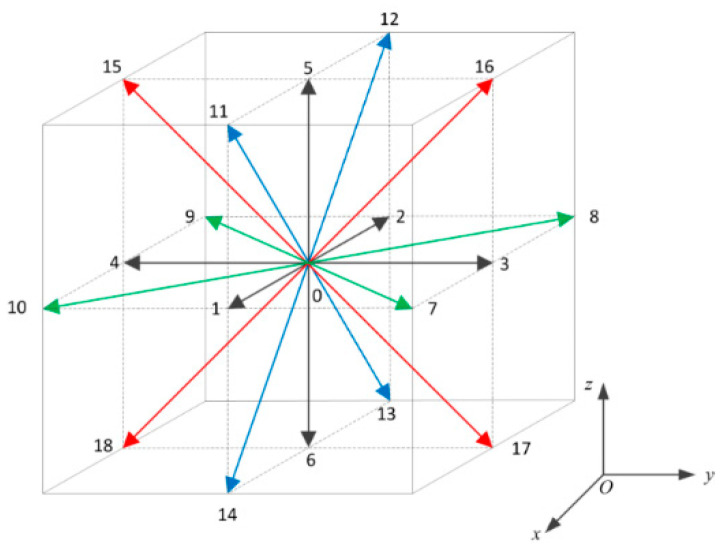
Discrete velocities of D3Q19 model.

**Figure 2 polymers-13-00133-f002:**
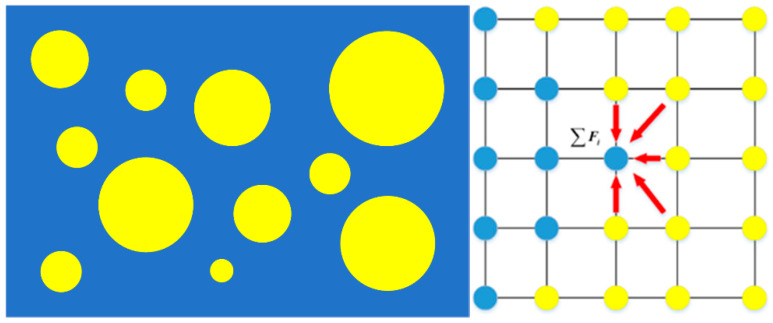
Pseudo-potential force between particles of droplets and the matrix.

**Figure 3 polymers-13-00133-f003:**
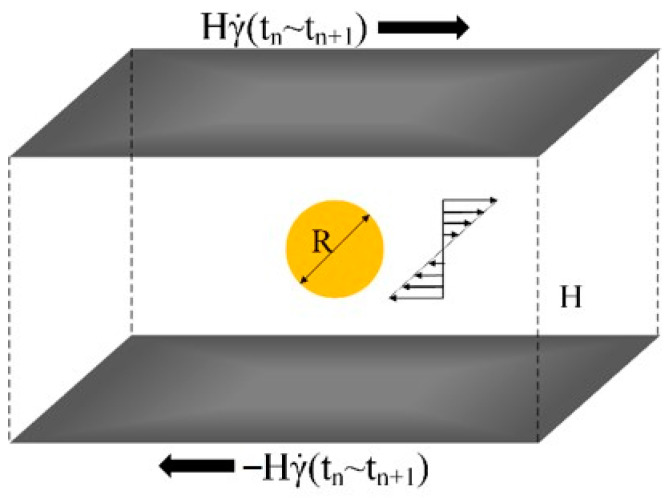
Set-up of a local model for droplet morphology evolution between *t* and *t* + Δ*t*.

**Figure 4 polymers-13-00133-f004:**
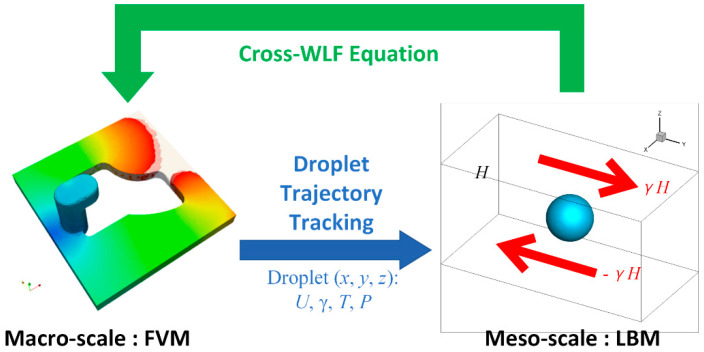
Coupling of macroscopic filling flow and mesoscopic morphology evolution.

**Figure 5 polymers-13-00133-f005:**
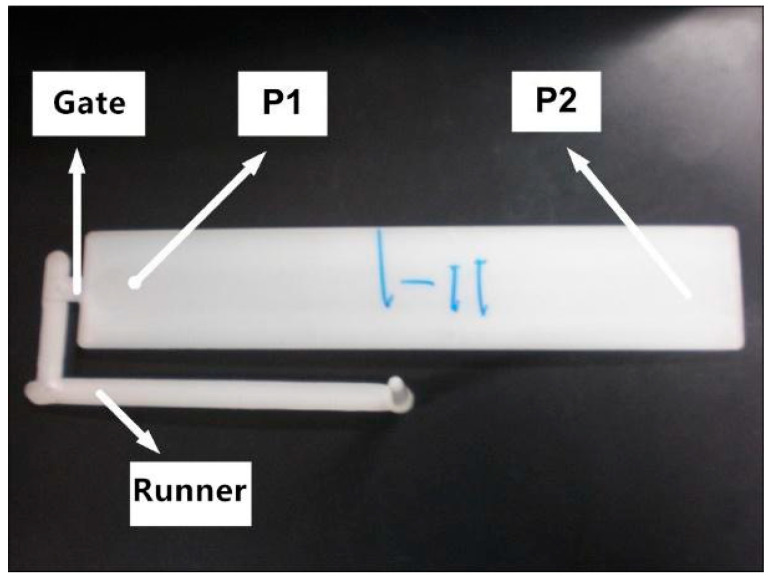
Two sensors capturing cavity pressure at P1 and P2.

**Figure 6 polymers-13-00133-f006:**
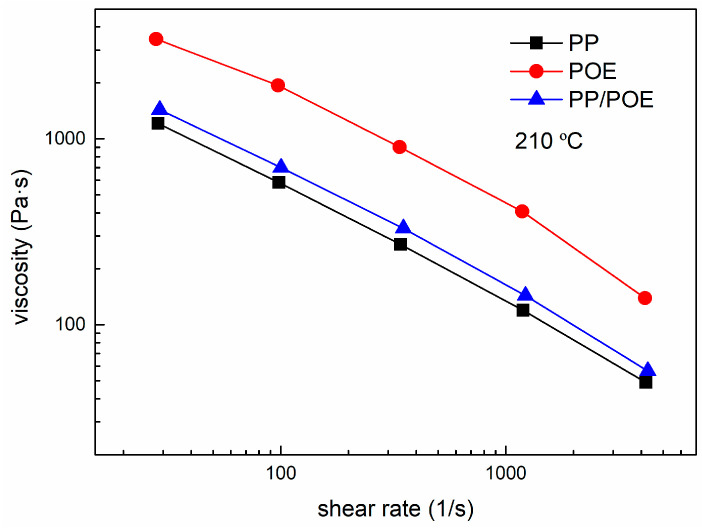
Shear viscosities of PP, POE and theirs blends at 210 °C.

**Figure 7 polymers-13-00133-f007:**
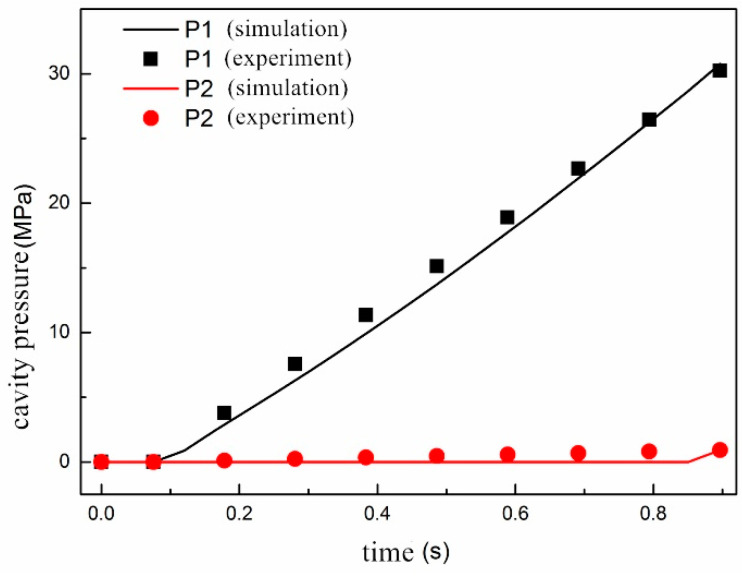
Comparison of cavity pressure variation at P1 and P2 over time.

**Figure 8 polymers-13-00133-f008:**
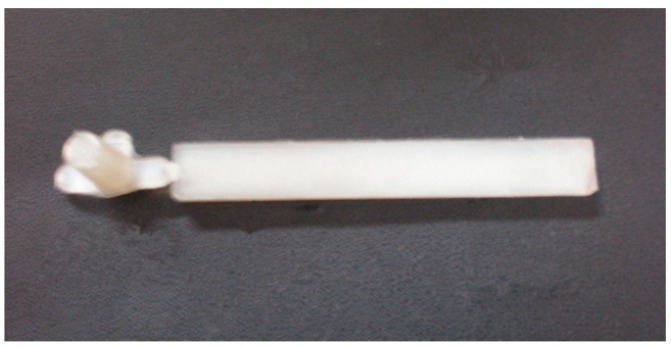
The molded part for droplet morphology experiment.

**Figure 9 polymers-13-00133-f009:**
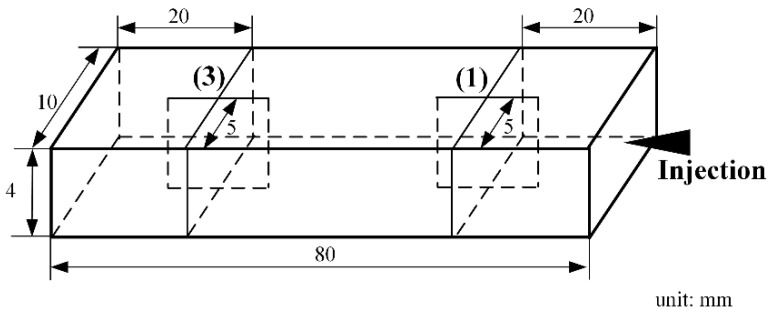
Details of sections on the part for the droplet morphology experiment.

**Figure 10 polymers-13-00133-f010:**
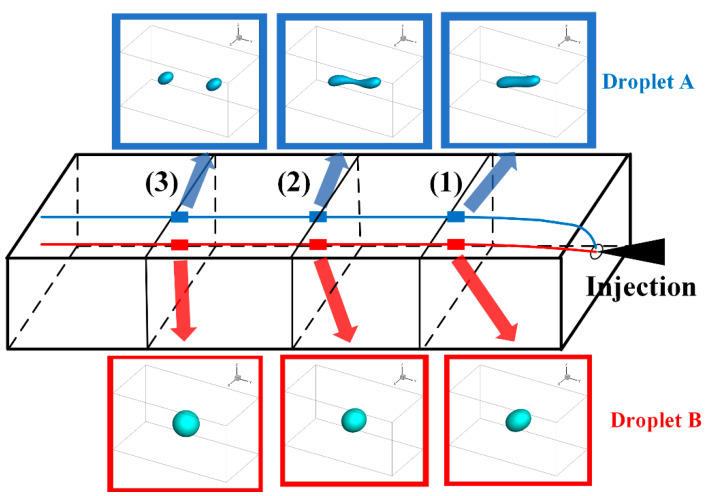
Trajectories and morphologies of the droplets A and B.

**Figure 11 polymers-13-00133-f011:**
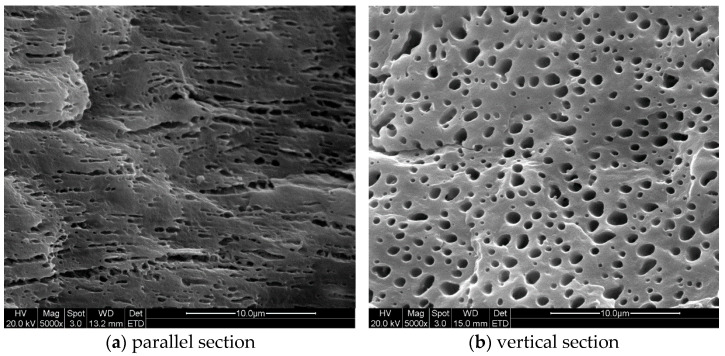
Morphologies of the droplets in shear layer at position (1) of case II part.

**Figure 12 polymers-13-00133-f012:**
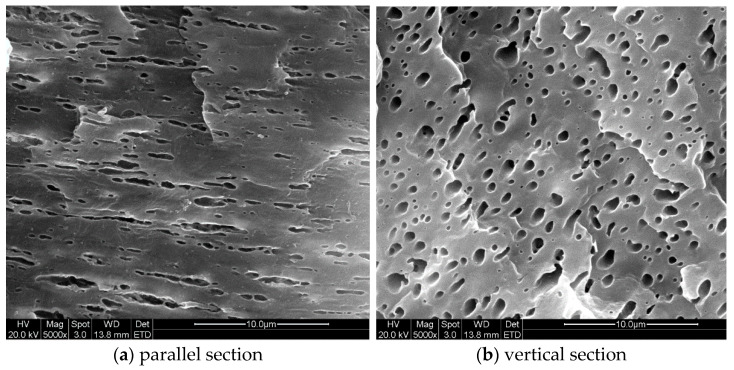
Morphologies of the droplets in shear layer at position (3) of case II part.

**Figure 13 polymers-13-00133-f013:**
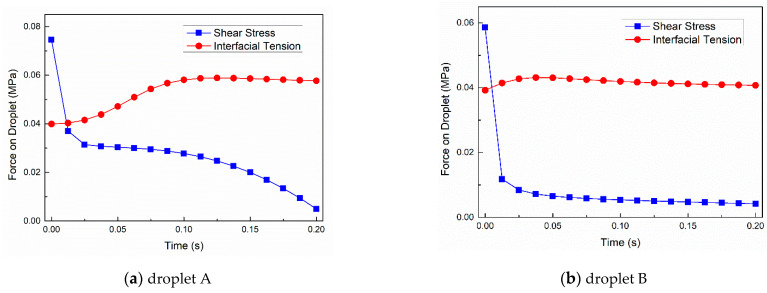
Shear stress and interfacial tension during mold filling.

**Figure 14 polymers-13-00133-f014:**
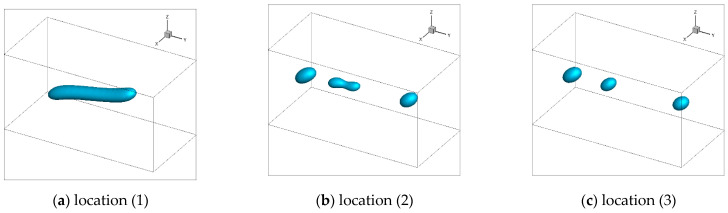
Simulated morphologies of droplet A under high injection rate.

**Figure 15 polymers-13-00133-f015:**
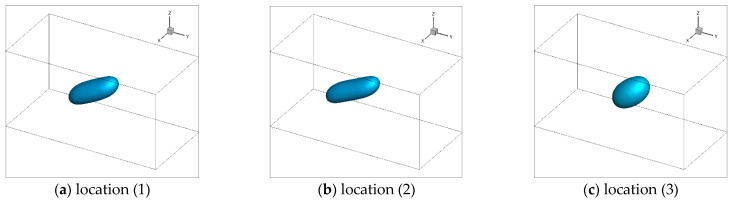
Simulated morphology of droplet A under low injection rate.

**Figure 16 polymers-13-00133-f016:**
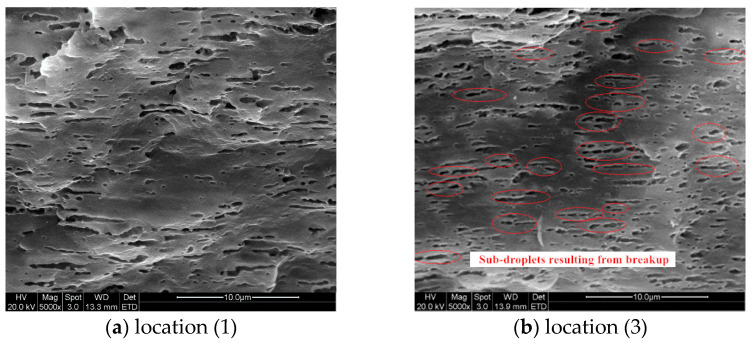
SEM images of the droplets in shear layer of case III part.

**Figure 17 polymers-13-00133-f017:**
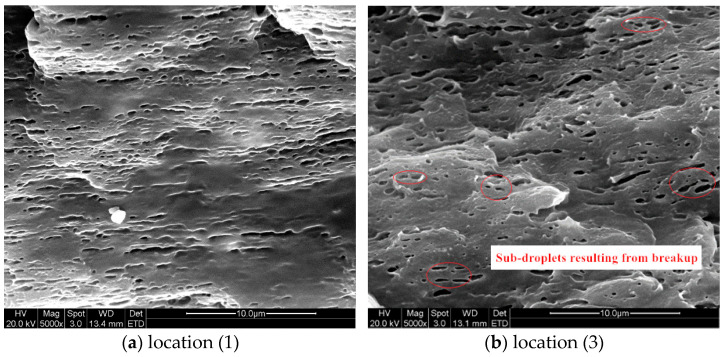
SEM images of the droplets in shear layer of case I part.

**Figure 18 polymers-13-00133-f018:**
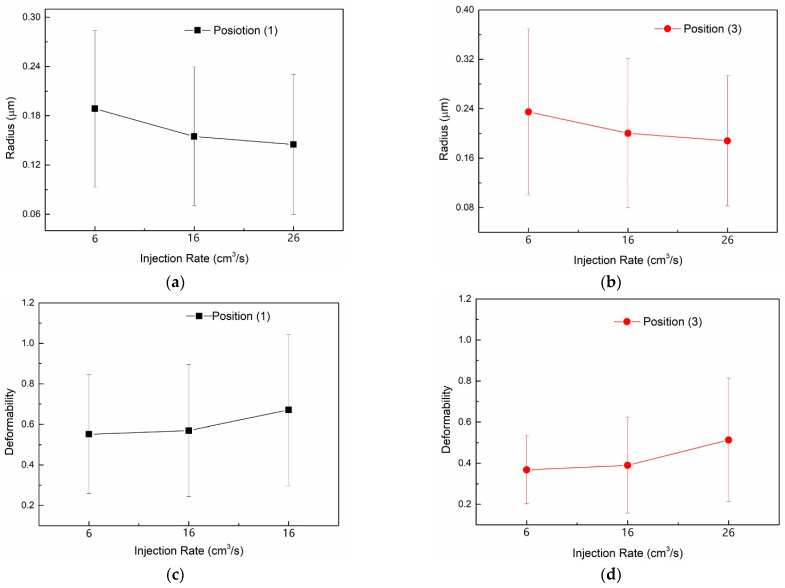
Statistics of droplets morphologies on the SEM images: equivalent radius of the droplets at Position 1 (**a**) and 3 (**b**); deformability of the droplets at Position 1 (**c**) and 3 (**d**).

**Table 1 polymers-13-00133-t001:** Variables of the Governing Equations.

Variable	Meaning	Variable	Meaning
***u***	velocity	*c_p_*	specific heat
*P*	pressure	*k*	thermal conductivity
*T*	temperature	*μ*	viscosity
*t*	time	*ρ*	density
g	gravity	$\dot{\Phi }$	heat source

**Table 2 polymers-13-00133-t002:** Physical meanings of the symbols in the generalized transport equation.

Equation	*ϕ*	Λ	Γ	*Q_ϕ_*
Continuity	1	1	0	0
Momentum	u	ρ	η	$\nabla \cdot \left[\eta {\left(\nabla u\right)}^{T}\right]-\nabla p$
Energy	T	$\rho c_{p}$	k	$2\eta \dot{\gamma }:\dot{\gamma }$

**Table 3 polymers-13-00133-t003:** Processing parameters in of the cavity pressure experiment.

Processing Conditions	Parameters
Injection rate (cm^3^/s)	25
Mold temperature (°C)	20
Packing pressure (MPa)	45
Packing time (s)	10
Cooling time (s)	10
Melt temperature (°C)	225

**Table 4 polymers-13-00133-t004:** Physical properties of PP and POE.

Parameters	PP	POE
Melt density (g/cm^3^)	0.738	0.776
Thermal conductivity (J/(kg∙°C))	2755	2380
Heat capacity (w/(m∙°C))	0.173	0.236
Melt flow rate (g/10 min)	3.5	1.0
Melt index (10 g/min)	3.6	0.50

**Table 5 polymers-13-00133-t005:** Cross-WLF coefficients of PP/POE blend.

$\dot{\tau }$ (Pa)	$\tilde{n}$	A1	A2 (K)	D1 (Pa·s)	D2 (K)	D3 (K/Pa)
40,800.845	0.29	16.23	220.15	4.25 × 107	259	0

**Table 6 polymers-13-00133-t006:** Processing conditions for the droplet morphology experiment.

	Case I	Case II	Case III
Injection rate (cm^3^/s)	6	16	26
Injection temperature (°C)	225	225	225
Mold wall temperature (°C)	20	20	20
Packing time (s)	10	10	10
Cooling time (s)	10	10	10

## Data Availability

Data available on request due to restrictions eg privacy or ethical. The data presented in this study are available on request from the corresponding author. The data are not publicly available due to research conditions.

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
