# Peer review of "A Novel Multiscale Methodology for Simulating Droplet Morphology Evolution during Injection Molding of Polymer Blends"

_polymers, 2020, doi:10.3390/polym13010133_

Round 1
Reviewer 1 Report
This study investigates morphology development in the injection molding of a polymer blend experimentally and numerically. Particularly in the numerical simulation, the authors presented a multi-scale method that combines macroscale and mesoscale simulations, where the droplet trajectory and the shear rate computed from the macroscale simulation is employed for the mesoscale simulation. The manuscript is well written in general, and it could be of interest to the readership of the Polymers journal. Though, I want to give one comment as follows.
In the injection molding filling stage, the flow near the melt-front region is characterized by a fountain flow where the polymer melt in the core region can travel to near the wall. This fountain flow can affect the morphology of the product, especially near the wall. I believe the author's macroscale simulation tool can predict the fountain flow because their numerical model is fully three-dimensional. In this regard, I want to recommend the authors to add some results and discussion regarding the fountain flow effect if possible, so that their manuscript can be much strengthened.
Author Response
Dear Editors and Reviewers:
Thank you for your letter and for the reviewers’ comments concerning our manuscript entitled “A Novel Multiscale Methodology for Simulating Droplet Morphology Evolution During Injection Molding of Polymer Blends” (ID: polymers-1028157). Those comments are all valuable and very helpful for revising and improving our paper, as well as the important guiding significance to our researches. We have studied comments carefully and have made correction which we hope meet with approval. Revised portion are marked in red in the paper. The main corrections in the paper and the responds to the reviewer’s comments are as flowing:
Reviewer 1:
- “In the injection molding filling stage, the flow near the melt-front region is characterized by a fountain flow … In this regard, I want to recommend the authors to add some results and discussion regarding the fountain flow effect if possible, so that their manuscript can be much strengthened.”
Response: Thanks very much for the desirable advice from the reviewer! Fountain flow is a typical phenomenon of the mold-filling flow of polymer melt. It’s a worthwhile issue to study the morphology evolution of the droplet in the flow front and the macroscopic model of this paper is capable of doing that. However, the difficulty lies in the experimental characterization of the droplet morphology there. Right now we’re furthering the study to overcome this problem. We have briefed this issue in the Conclusion and Outlook section in the end of the paper and we earnestly welcome other researchers taking interests in this field.
We tried our best to improve the manuscript and made some changes in the manuscript. These changes will not influence the content and framework of the paper. All the changes have been annotated in revised paper.
We appreciate for Editors/Reviewers’ warm work earnestly, and hope that the correction will meet with approval.
Once again, thank you very much for your comments and suggestions.

Reviewer 2 Report
The present paper reports a novel multiscale model for studying droplet morphology evolution of polymer blends during mold filling process using the finite volume method and lattice Boltzmann method. The paper is well presented and the results are confirmed by reported model and experimentation. The Authors are encouraged to address the following comments.
- In the introduction section, the author should put better in evidence from literature the direct relation between the droplet evolution morphology and the microstructure.
- In the experimental section, how were the process parameters defined? On which considerations was the blend fraction fixed? Which was the molding machine used?
- Line 313: could the authors report the “smaller dimensions” of the second type of samples?
- Line 394: what about the standard deviation of statistics in Fig. 18?
- In fig. 18 I think that the abscissa should reports fixed values. Why is there a graduate scale? In correspondence of each column what is the injection rate value?
- The authors are invited to report the definitions of “average radius” (I suppose the radius of the circle with the same area of the droplet) and deformability (how was it calculated?).
- Looking at reference, I found 5-6 papers on total of 42 that are from the last 5 years. Do you think that, at present, this is an argument worth of interest?
- In the paper, the aspect of “blend”, reported since the title, was not discussed thoroughly and in particular the effect of the POE fraction on droplet morphology.
English and typos
- In the manuscript, the form “it’s” is used many times instead of “it is” that I suggest for a written text.
- Line 108: please remove “The rest of”
- Section 3.2.3 Missing the initial word of the sentence.
Author Response
Dear Editors and Reviewers:
Thank you for your letter and for the reviewers’ comments concerning our manuscript entitled “A Novel Multiscale Methodology for Simulating Droplet Morphology Evolution During Injection Molding of Polymer Blends” (ID: polymers-1028157). Those comments are all valuable and very helpful for revising and improving our paper, as well as the important guiding significance to our researches. We have studied comments carefully and have made correction which we hope meet with approval. Revised portion are marked in red in the paper. The main corrections in the paper and the responds to the reviewer’s comments are as flowing:
Reviewer 2:
- “In the introduction section, the author should put better in evidence from literature the direct relation between the droplet evolution morphology and the microstructure.”
Response: Thanks for the suggestion. The reference concerning the relationship between the droplet morphology evolution and the microstructure has been supplemented in the introduction section.
- “In the experimental section, how were the process parameters defined?”
Response: The processing parameters were defined mainly based on the experience of the injection molding practice according to the geometry, size and polymer of the molded product, referring to the parameters provided by Moldflow Synergy meanwhile. This was added in the experiment section.
- On which considerations was the blend fraction fixed?
Response: There are two reasons for the fixed fraction of PP/POE blend. Firstly, the research of this paper is carried out under the background of the manufacturing of polymer blends product. Trough mechanical testing of the products of various fraction, it is found that the blend of this fraction demonstrates the optimal mechanical performance and formability. Secondly, the fraction of components decides the morphology of the blends and the matrix-droplets, the study object of this paper, other than co-continuous structure will occur when the fraction is low than a critical level. This consideration has been made clear in the revised manuscript.
- “Which was the molding machine used?”
The molding machine used in this paper is the Type HTL-90-F5B injection machine (produced by Ningbo Haitai Plastic Machinery Co., Ltd.). The machine information has been stated in the paper thanks for the reminder of the reviewer.
- “Line 313: could the authors report the “smaller dimensions” of the second type of samples?”
Response: The size of the second type of samples is 80×10×4 mm3. It’s my fault to fail detailing this important data. Thanks a lot!
- “Line 394: what about the standard deviation of statistics in Fig. 18?
Response: The Fig.18 has been redrawn so that the standard deviation of statistics is included now. Thanks for helping us improve the Figure!
In fig. 18 I think that the abscissa should reports fixed values. Why is there a graduate scale? In correspondence of each column what is the injection rate value?
Response: That is a mistake of expression and does not exist in the redrawn Fig. 18.
The authors are invited to report the definitions of “average radius” (I suppose the radius of the circle with the same area of the droplet) and deformability (how was it calculated?).”
Response: Thanks again for the reviewer’s reminder. That’s necessary to tell clearly that how were them calculated. This paper assumes the droplets to be ellipsoid. With the help of a powerful software (Nano Measurer), the long axis and short axis of the droplets in the SEM imaged was measured and analyzed using the software Origin Lab. For clarity, the definition formula of the deformed droplets was added in this paper.
- “Looking at reference, I found 5-6 papers on total of 42 that are from the last 5 years. Do you think that, at present, this is an argument worth of interest?”
Response: As elaborated in the Introduction, this field is certainly significant and interesting. However, the research of this field encountered a bottleneck caused by the multiscale characteristic of polymer blends. That is also why the author constantly devote to this research. Thanks for the suggestions from the reviewer, the author has retrieved several reviews and articles concerning this issue in recent years.
- “In the paper, the aspect of “blend”, reported since the title, was not discussed thoroughly and in particular the effect of the POE fraction on droplet morphology.”
Response: This reason is similar to that of Response 3. The fraction of POE does have obvious impact on the morphology of polymer blends and was being further studied in our team. We also welcome other academics taking interest in this issue.
“English and typos”
In the manuscript, the form “it’s” is used many times instead of “it is” that I suggest for a written text.
Line 108: please remove “The rest of”
Section 3.2.3 Missing the initial word of the sentence.
Response: Thanks for your careful reviewing. All the mistakes you mentioned above have been fixed.
We tried our best to improve the manuscript and made some changes in the manuscript. These changes will not influence the content and framework of the paper. All the changes have been annotated in revised paper.
We appreciate for Editors/Reviewers’ warm work earnestly, and hope that the correction will meet with approval.
Once again, thank you very much for your comments and suggestions.
